# Correlations among Psychological Resilience, Cognitive Fusion, and Depressed Emotions in Patients with Depression

**DOI:** 10.3390/bs13020100

**Published:** 2023-01-25

**Authors:** Ning Chen, Juzhe Xi, Xiwang Fan

**Affiliations:** 1Clinical Research Center for Mental Disorders, Shanghai Pudong New Area Mental Health Center, School of Medicine, Tongji University, Shanghai 200124, China; 2College of Arts, Humanities and Social Sciences, University of Edinburgh, Edinburgh EH8 9YL, UK; 3Shanghai Key Laboratory of Mental Health and Psychological Crisis Intervention, Affiliated Mental Health Center (ECNU), School of Psychology and Cognitive Science, East China Normal University, Shanghai 200062, China

**Keywords:** psychological resilience, cognitive fusion, depressed emotions, depressive patients, depression, anxiety

## Abstract

*Background:* More than 300 million people worldwide suffer from depression, which is a significant contributor to the global burden of disease. This study investigated the factors influencing psychological resilience and cognitive fusion in patients with depression and the relationships of psychological resilience and cognitive fusion with depression. *Methods:* This study enrolled 172 participants (65.8% of them were female). Psychological resilience, cognitive fusion, and depression were assessed with the psychological resilience scale, the cognitive fusion questionnaire, the Hamilton Anxiety Scale (HAMA), Hamilton Depression Scale (HAMD), and Zung Self-rating Depression Scale (SDS), respectively. Furthermore, the relationships of psychological resilience and cognitive fusion with depression were investigated. *Results:* The psychological resilience and cognitive fusion scores of patients with depression varied significantly among different education levels, and HAMA, HAMD, and SDS scores were significantly negatively correlated with psychological resilience but positively correlated with cognitive fusion. *Conclusions:* Depression levels in patients with depression are closely related to psychological resilience and cognitive fusion. Therefore, anxiety and depression could be alleviated by improving the psychological resilience or reducing the cognitive fusion of patients with depression.

## 1. Introduction

The World Health Organization estimates that more than 300 million people suffer from depression (major depressive disorder) worldwide, making it a significant contributor to the global burden of disease. The global average annual prevalence of depression is approximately 6%, and the lifetime prevalence of depression is 15% to 18% [1,2]. Since the COVID-19 pandemic began, the incidence of depression has more than tripled compared with the historical incidence [3]. The main clinical symptoms of depression are low mood, decreased interest, and unresponsiveness, and it is one of the most common mental disorders [4].

Despite an upsurge in research into depression in recent years, the cause and pathogenesis of depression are still unknown. Researchers generally agree that a combination of biopsychosocial and physiological factors causes depression and that cognition or thinking are also key influences on people’s moods [5]. Therefore, the study of cognition [6] or thinking related to negative emotions in patients with depression is necessary to explore the pathogenesis of depression. With the rise of the third generation of cognitive-behavioral therapy represented by acceptance and commitment therapy (ACT), research on the cognitive level has expanded. This perspective focuses on patients’ mental flexibility, acknowledges the universality of pain, requires patients to accept rather than resist pain and bad experiences, and emphasizes the role of mindfulness [7].

Cognitive fusion, an important theoretical model of ACT, is an essential factor in the formation and maintenance of many psychological disorders [8,9]. Cognitive fusion indicates that patients interpret their thoughts and emotions as factual and genuine, exacerbating patients’ negative psychological emotions [6,10,11]. When patients experience pathological cognitive fusion, their psychological flexibility is reduced, and psychological problems occur. Many studies have confirmed the association of psychological resilience with depression. For example, in an experiment that analyzed the relationship between depression and psychological flexibility in Korean conscripts, researchers found that psychological flexibility had a negative effect on depression, particularly in the dimension of resilience [11]. Therefore, this study investigated whether improving psychological flexibility could reduce depression symptoms in patients with depression.

Research on the relationship between cognitive fusion and mental health has focused on negative affect, psychological distress, anxiety disorders, and depression. For example, it has been reported that cognitive fusion is one of the critical factors contributing to psychological disorders [12]. Scholars have reported similar results about the differences in cognitive fusion between gender and age. Nevertheless, in research related to age and gender differences, some scholars have found negligible associations between psychological flexibility, gender, and age [13]. However, studies on depression have revealed differences in the prevalence and severity of symptoms between males and females with depression, and the gender differences in depression increase with age [14,15,16]. Therefore, investigating whether gender and age differences exist in the psychological flexibility and cognitive fusion of patients with depression is vital to examine the correlations between depression, psychological flexibility, and cognitive fusion.

The present study used a sample of Chinese patients with depression to develop a psychopathological model exploring the relationship between psychological resilience, cognitive fusion, and depression, providing a theoretical reference for clinicians to develop practical measures to predict depression and alleviate its symptoms.

Hypotheses: (1) Demographic differences exist between cognitive fusion and psychological flexibility scores in the clinical group of depression, age, and gender; (2) Cognitive fusion and psychological flexibility are significantly correlated with depression; (3) Cognitive fusion positively predicts depression levels in patients with depression; (4) Psychological flexibility negatively predicts depression levels in patients with depression.

## 2. Methods

### 2.1. Participants

Patients with depression treated in Shanghai Pudong Mental Health Center from March 2020 to March 2022 were selected. A trained senior psychiatrist confirmed the diagnosis of major depression according to the Diagnostic and Statistical Manual of Mental Disorders, Fourth Edition (DSM-IV). The inclusion criteria were as follows: (1) no previous episodes or treatment; (2) age 10–60 years, regardless of gender; (3) Hamilton Depression Scale score (HAMD, 17 items) ≥8; (4) Han nationality; (5) dextral; (6) junior high school education or above; (7) willingness to voluntarily participate in the study and sign the informed consent form; and (8) ability to communicate smoothly with the study staff.

The exclusion criteria were as follows: (1) organic psychiatric disorders, schizophrenia, and mood disorders; (2) visual or auditory disorders; and (3) drug or alcohol dependence, pregnancy, or lactation.

In total, 172 participants were recruited; 16 were excluded according to the above criteria, and 156 patients with depression were enrolled after obtaining informed consent. The patients included 50 males and 106 females, with a mean age of approximately 26 years and a mean educational level of high school education.

### 2.2. Research Tools

Self-compiled Questionnaire of General Situation: The general situation questionnaire included age, sex, and education.

Psychological Resilience Scale (the 14-Item Resilience Scale, RS-14): The Chinese version of the psychological resilience scale developed by Ni Qianyu and Tian Jun (2013) is a 14-item short form of the RS scale with a total score of 14–98. A higher score indicates greater psychological resilience. The Cronbach’s α coefficient of the scale is 0.928, showing good reliability and validity.

Cognitive Fusion Questionnaire (CFQ): The Chinese version of the cognitive fusion questionnaire (CFQ) was developed by Zhang Weichen et al. (2004). The Cronbach’s α coefficient of the scale is 0.92 and the retargeting validity is 0.67. The questionnaire consists of 13 items and is divided into two sub-questionnaires: the Cognitive Fusion Questionnaire—Fusion (CFQ-F) and the Cognitive Fusion Questionnaire—Defusion (CFQ-D). The CFQ-F contains items 1, 2, 4, 5, 7, 8, 10, 11, and 13, and the CFQ-D contains items 3, 6, 9, and 12. Each item is scored on a 7-point scale: 1 = clearly does not fit; 2 = does not fit; 3 = somewhat does not fit; 4 = in the middle; 5 = somewhat fits; 6 = fits; 7 = clearly fits. Higher CFQ-F scores indicate higher degrees of cognitive integration; CFQ-D is the reverse description of cognitive integration, and higher scores indicate lower degrees of cognitive integration and higher degrees of cognitive dissociation.

Hamilton Anxiety Scale (HAMA): The Hamilton anxiety scale (HAMA), developed in 1960, is used clinically to assess the severity of the participants’ anxiety disorders. The scale has 14 items, and each is scored from 0 to 4. The scale contains two factors: somatic anxiety and mental anxiety. A higher total score indicates more serious anxiety (Zimmerman et al., 2020).

Hamilton Depression Scale (HAMD): A 17-item version of the HAMD (Zimmerman et al., 2013) was adopted in this study, which is divided into five factors: anxiety/somatization, weight, cognitive impairment, block, and sleep disorders. Each item is scored from 0 to 4, and a higher score indicates more severe depression.

Self-Rating Depression Scale (SDS): The depression level of patients was assessed by the Zung self-rating depression scale (SDS). A total score ≥50 is classified as depression-positive. Higher scores indicate a higher degree of depression. The Cronbach’s α coefficient of the scale is 0.885, the content validity is 0.826, and the internal consistency of the scale is good.

### 2.3. Procedures

A psychiatrist diagnosed the participants according to the DSM-IV diagnostic criteria. An assistant researcher informed the participants and the guardians of the minors about the purpose of the research. After informed consent was obtained, the assistant investigator evaluated the participants according to the HAMD, HAMA, and SDS items and instructed the participants to complete the General Demographics Questionnaire, RS-14, and CFQ to ensure that participants under 18 years of age understood the items.

### 2.4. Statistical Analysis

Data analysis was conducted using SPSS 26.0 software. The means and standard deviations of the RS-14 and CFQ were calculated; *t*-test and analysis of variance (ANOVA) were used to compare the scores of the HAMD, HAMA, SDS, CFQ, and RS-14 and its three different dimensions in patients with depression with different demographic characteristics; Pearson product–moment correlation was used to test the correlation between different variables; and cognitive fusion and psychological resilience were considered as independent variables to conduct multiple linear stepwise regression for patients’ anxiety and depression levels to explore the prediction extent of psychological resilience and cognitive fusion. A two-tailed *p*-value of <0.05 was considered statistically significant.

## 3. Results

### 3.1. Descriptive Statistics of Variables

We use number 1 for males and number 2 for females, and this study included 50 males and 106 females. The education level of participates was coded as follows: “1” for junior high school and below; “2” for high school or equivalent-educated; “3” for undergraduate degree; and “4” for postgraduate degree or above. The mean values of each variable are shown in Table 1, and most of them maintained at a moderate level.

### 3.2. Different Variables among Different Demographic Characteristics of Patients with Depression

In Table 2, the one-way ANOVA results showed significant differences in psychological resilience and cognitive fusion scores among patients with depression of different ages. In addition, independent sample *t*-tests revealed that the psychological resilience and cognitive fusion scores of patients with depression did not show significant differences in terms of gender. However, the ANOVA results showed significant differences in psychological resilience (*p* < 0.05) and cognitive fusion (*p* < 0.01) scores among patients with depression with different education levels.

### 3.3. Correlation Analysis Results of Depression, Psychological Resilience, and Cognitive Fusion

Pearson’s product–moment correlations were used to calculate the relationship among psychological resilience and its three dimensions: cognitive fusion, anxiety, and depression. The results in Table 3 showed that psychological resilience was negatively correlated with HAMD, HAMA, and SDS (*p* < 0.001); by contrast, cognitive fusion was positively correlated with HAMD, HAMA, and SDS (*p* < 0.001). A significant negative correlation was found between cognitive fusion and psychological resilience (*p* < 0.001).

### 3.4. Results of Multiple Linear Stepwise Regression Analysis of Psychological Resilience, Cognitive Fusion, Depression, and Anxiety

The normality test was first performed and all satisfied normality. Multiple linear stepwise regression analysis was performed with HAMA, HAMD, and SDS criteria scores of the sample as dependent variables; psychological resilience and cognitive fusion scores as independent variables; and gender, age, and education level as control variables. As the results showed in Table 4, psychological resilience negatively predicted HAMA, HAMD, and SDS, while cognitive fusion positively predicted HAMA, HAMD, and SDS. The coefficient of variance inflation for each regression equation is not greater than 5, indicating that there is no multicollinearity between the independent variables within the regression equation.

## 4. Discussion

In this study, 172 patients with clinical depression were recruited to study the relationships between psychological resilience, cognitive fusion, and depression. The results showed that different levels of education were influential factors for psychological resilience and cognitive fusion in patients with depression. People with more years of education may be able to judge what is happening more rationally and cope in a reasonable manner. Additionally, no significant differences were observed in cognitive fusion and psychological resilience in patients with depression by gender and age of onset, which is consistent with the findings of previous related studies [17,18]. Interestingly, unlike previous findings, females’ average psychological resilience scores were higher than males’.

The results of the cognitive fusion and depression correlation analysis showed that cognitive fusion was significantly and positively associated with depression levels, which is consistent with the findings of Bardeen and Fergus [19]. This suggests that when negative thinking limitations influence emotions, adverse feelings may be further reinforced, leading to increased negative affect and depressive symptoms [20]. This is also largely consistent with the theoretical underpinnings of ACT, where cognitive integration is thought to exacerbate emotional distress when a person is unable to use alternatives flexibly [21,22].

By contrast, a significant negative correlation was observed between psychological resilience and depression levels. This study found that more severe depression levels were associated with lower psychological resilience scores. This may be because some patients have psychological problems such as anxiety, depression, and fear, which affect the adaptation to mental illness and lead to poor psychological resilience [23]. Additionally, the process is cyclically exacerbated, people with low psychological resilience are unable to adapt to stressful events and therefore tend to experience poorer mental health problems, like depression and anxiety [24].

In addition, cognitive fusion and psychological resilience were negatively correlated. Because of pathological cognitive fusion, patients appear to confuse thoughts with reality and dwell on negative thoughts and emotions. When ‘fused’, a person becomes less sensitive to immediate or genuine consequences and cognitive events begin to dominate behaviour and experience rather than other emotional or behavioral regulation [25]. Under the influence of negative thoughts, patients’ self-efficacy decreases and deviates from self-worth, leading to low psychological resilience and psychological disorders. A higher degree of cognitive fusion in patients with depression is associated with lower psychological resilience and a greater likelihood of developing or exacerbating psychological problems.

This study investigated the relationships between cognitive integration, psychological resilience, and depression in patients with depression and found that cognitive integration was positively associated with symptoms of depression. By contrast, psychological resilience had a negative predictive effect on symptoms of depression, providing theoretical guidance for the diagnosis and treatment of depressed patients. However, this study has several limitations: (1) the patients included in this study were all recruited from the same psychiatric center, and thus the diversity of the sample is limited to some extent; (2) this is a cross-sectional and correlational study, which limits the ability to infer causality, and further longitudinal studies should be conducted to replicate and build on the results of this study; (3) the various dimensions of psychological resilience, such as resilience, strength, and optimism, may have different degrees of influence on depression, and this pathological model requires further refinement.

## 5. Conclusions

In summary, HAMA scores, HAMD scores, and SDS scores were negatively correlated with psychological resilience in this study, suggesting a close relationship between the degree of depression and psychological resilience in patients with depression. Higher levels of anxiety and depression tended to be accompanied by lower psychological resilience and a greater degree of cognitive fusion, exacerbating the level of anxiety or depression. This suggests that clinicians should improve patients’ psychological resilience and provide interventions to individuals with high cognitive fusion to avoid negative cognitions, for example, Mindfulness-based Stress Reduction [26] and ACT [27], which reduces cognitive fusion and can help patients to some extent to alleviate symptoms of anxiety and depression. Measures like social support, coping self-efficacy and improving physical health [28] should also be taken to help patients with low education levels improve their psychological resilience, and decrease anxiety and depression.

## Figures and Tables

**Table 1 behavsci-13-00100-t001:** Descriptive statistics of the variables (N = 156).

Variables	Mean	Standard Deviation	Variance
*Gender*	1.68	0.47	0.22
*Age*	26.76	10.55	111.28
*Education*	2.52	0.82	0.68
*Toughness*	19.28	8.24	67.83
*Strength*	14.24	4.74	22.48
*Optimism*	7.36	3.31	10.94
*Psychological resilience*	40.88	14.87	221.08
*Cognitive fusion*	47.71	10.59	112.15
*HAMA*	16.11	6.13	37.57
*HAMD*	22.67	7.10	50.39
*SDS*	54.03	10.58	111.86

**Table 2 behavsci-13-00100-t002:** Comparison of the scores for psychological resilience and cognitive fusion among patients with depression with different demographic characteristics (scores, x ± s).

Variables	N	*Psychological Resilience*			*Cognitive Fusion*		
		M ± SD	t/F	P	M ± SD	t/F	P
*Gender*			1.028	0.305		0.044	0.965
*Male*	50	42.66 ± 14.98			47.76 ± 11.33		
*Female*	106	47.76 ± 11.33			47.68 ± 10.28		
*Age*			1.530	0.209		0.117	0.950
*20 and below*	57	37.32 ± 13.08			49.35 ± 10.09		
*20–30*	42	40.29 ± 15.56			48.76 ± 9.99		
*30–40*	37	43.32 ± 15.23			48.57 ± 9.75		
*40 and above*	20	47.75 ± 15.42			39.20 ± 11.46		
*Education*			2.981 *	0.033		5.405 **	0.001
*Junior High and below*	22	41.86 ± 15.13			46.45 ± 10.68		
*High school Diploma*	42	36.86 ± 12.06			47.88 ± 10.63		
*University Diploma*	81	42.16 ± 15.29			47.93 ± 10.41		
*Master and above*	11	44.82 ± 19.47			47.91 ± 12.82		

Note: ** significant correlation at the 1% level (two-tailed); * significant correlation at the 5% level (two-tailed).

**Table 3 behavsci-13-00100-t003:** Correlation analysis results of depression, anxiety, cognitive fusion, and psychological resilience and its three dimensions.

Variables	*RS-14*	*CFQ*	*HAMD*	*HAMA*	*SDS*
*RS-14*	1.000				
*CFQ*	−0.367 **	1.000			
*HAMD*	−0.376 **	0.414 **	1.000		
*HAMA*	−0.306 **	0.361 **	0.729 **	1.000	
*SDS*	−0.552 **	0.585 **	0.697 **	0.565 **	1.000

Note: RS-14, the 14-Item resilience scale; CFQ, cognitive fusion questionnaire; HAMD, Hamilton depression scale; HAMA, Hamilton anxiety scale; SDS, self-rating depression scale; ** significant correlation at the 1% level (two-tailed).

**Table 4 behavsci-13-00100-t004:** Results of regression analysis of psychological resilience, cognitive fusion, anxiety, and depression.

Dependent Variables	Predictor Variables	β	t	P	R^2^	Adjust R^2^
*HAMA*	*RS-14*	−0.083	−2.525	0.013	0.165	0.154
	*CFQ*	0.166	3.614	0.000		
*HAMD*	*RS-14*	−0.124	−3.400	0.001	0.230	0.219
	*CFQ*	0.214	4.178	0.000		
*SDS*	*RS-14*	−0.277	−6.18	0.000	0.474	0.467
	*CFQ*	0.441	7.003	0.000		

## Data Availability

The raw data supporting the conclusions of this article will be made available by the authors, without undue reservation.

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
