# Peer review of "Correlations among Psychological Resilience, Cognitive Fusion, and Depressed Emotions in Patients with Depression"

_behavsci, 2023, doi:10.3390/bs13020100_

Round 1
Reviewer 1 Report
On abstract I suggest that the summary should include the number of the participant on whom the analysis was done. I suggest to put the age limits and the percentage by gender.
On 3.1. Descriptive statistics of variables, lines 143-150
Methodology
The results of the study are based on a small sample of 156 patients with depression. Two classical statistical tests were used, t-test and ANOVA which for a correct interpretation require the satisfaction of assumptions such as normality, homogeneity, etc. The study does not mention whether these assumptions have been satisfactorily verified and therefore the results obtained are not reliable even misleading. Also, the verification of the assumptions for the linear regression model is not presented which affects the confidence in the presented results. Probably, taking into account the volume of the sample and the possible deviation from the statistical assumptions, the use of nonparametric methods would have been more appropriate for this study.
The text that introduces table no. 1 should not be like that with the repetition of some numbers that are already in the table. Only a short sentence is enough. Maybe only with low, medium, high correlation strength
In the text that introduces table 2, the numbers that are already in the table should not be included. The reader can read them from there in the table. It will only be said what the differences were, the correlations.
This aspect is also valid for table 3, 4. Numbers that are already in the table must not be written in the text that introduces the table.
Row 204-205
These number (figures) 40.88 ± 14.87 must be removed. These are already put in the tables from the results
The discussion section should be expanded on each scale, the hypothesis, with more comparisons with the specialized literature.
The conclusions section should be redone, more concretely, applied with suggestions. Now in the conclusions section, are repeated what is discussion.
Author Response
Dear reviewer,
Re: Manuscript ID: 2079556 and Title: Correlations among psychological resilience, cognitive fusion, and depressed emotions in patients with depression
Thank you for your letter and the reviewers’ comments concerning our manuscript entitled “Correlations among psychological resilience, cognitive fusion, and depressed emotions in patients with depression” (2079556). Those comments are valuable and very helpful. We have read through the comments carefully and have made corrections. Based on the instructions provided in your letter, we uploaded the file of the revised manuscript. The responses to your comments are presented following.
Q1. On abstract I suggest that the summary…
Response: We are grateful for the suggestion. To be more clear and in accordance with the reviewer's concerns, we have added a description of the number of participants and the proportion of gender.
Q2. Moderate English changes required…
Response: We apologize for the language problems in the original manuscript. The language presentation will be improved with assistance from a native English speaker with an appropriate research background.
Q3. The text that introduces table 1,2,3,4...,
Response: We agree with the comment and rewrote the sentence in the revised manuscript. And we removed those numbers already put in the table.
Q4. The discussion section should be expanded on each scale, and the hypothesis, with more comparisons with the specialized literature.
Response: We are grateful for the suggestion. As suggested by the reviewer, we have added more details on each aspect.
Q5. The conclusions section should be redone, more concretely, and applied with suggestions. Now in the conclusions section, are repeated what is discussion.
We agree with the comment and rewrote the sentence in the revised manuscript.
Q6. The study does not mention whether these assumptions have been satisfactorily verified...
We are grateful for the suggestion. We performed a stepwise regression analysis, the results of which are shown in the figure, and we also verified the normality of the data, among other things, indicating that it met the conditions. (Sorry, I've downloaded the P-P diagrams for each variable but can't find where to insert them.)
The specific changes are attached and I have marked them on the pdf.
We would love to thank you for allowing us to resubmit a revised copy of the manuscript and we highly appreciate your time and consideration.
Sincerely.
Fan X.

Reviewer 2 Report
You are not looking at cause and effect but just correlation. It would be nice if you could be more detailed in describing in behavior terms some of your suggestions. You recommend "help patients with low education levels improve their psychological resilience, reduce their negative cognition, and decrease anxiety and depression." Operationally define how one can go about doing that in actions (behavior) as just stating terms that are mentalistic and do not lead to a course of positive action.
Author Response
Dear reviewer,
Re: Manuscript ID: 2079556 and Title: Correlations among psychological resilience, cognitive fusion, and depressed emotions in patients with depression
Thank you for your letter and the reviewers’ comments concerning our manuscript entitled “Correlations among psychological resilience, cognitive fusion, and depressed emotions in patients with depression” (2079556). Those comments are valuable and very helpful. We have read through the comments carefully and have made corrections. Based on the instructions provided in your letter, we uploaded the file of the revised manuscript. The responses to your comments are presented following.
Q1. It would be nice if you could be more detailed…
Response: We are grateful for the suggestion. To be more clear and in accordance with the reviewer's concerns, we have added a more detailed interpretation in the end( I have marked the details in the attached pdf).
We would love to thank you for allowing us to resubmit a revised copy of the manuscript and we highly appreciate your time and consideration.
Sincerely.
Fan X
